# Effectiveness of a videoconferencing group-based dyad acceptance and commitment therapy on the quality of life of chronic heart failure patients and their family caregivers: A study protocol for a randomized controlled trial

Xuelin Zhang⬩, Grace W. K. Ho, Yim Wah Mak⬩*

School of Nursing, The Hong Kong Polytechnic University, Hong Kong, China

* yw.mak@polyu.edu.hk

## Abstract

### Background

Chronic heart failure (CHF) poses a significant burden on both patients and their family caregivers (FCs), as it is associated with psychological distress and impaired quality of life (QOL). Acceptance and Commitment Therapy (ACT) supports QOL by focusing on value living and facilitates acceptance of psychological difficulties by cultivating psychological flexibility. A protocol is presented that evaluates the effectiveness of a dyad ACT-based intervention delivered via smartphone on QOL and other related health outcomes compared with CHF education only.

### Methods

This is a single-center, two-armed, single-blinded (rater), randomized controlled trial (RCT). One hundred and sixty dyads of CHF patients and their primary FCs will be recruited from the Cardiology Department of a hospital in China. The dyads will be stratified block randomized to either the intervention group experiencing the ACT-based intervention or the control group receiving CHF education only. Both groups will meet two hours per week for four consecutive weeks in videoconferencing sessions over smartphone. The primary outcomes are the QOL of patients and their FCs. Secondary outcomes include psychological flexibility, psychological symptoms, self-care behavior, and other related outcomes. All outcomes will be measured by blinded outcome assessors at baseline, immediately post-intervention, and at the three-month follow-up. Multilevel modeling will be conducted to assess the effects of the intervention.

### Discussion

This study is the first to adopt an ACT-based intervention for CHF patient-caregiver dyads delivered in groups via smartphone. If effective and feasible, the intervention strategy and

**Data Availability Statement:** No datasets were generated or analysed during the current study. All

relevant data from this study will be made available upon study completion.

**Funding:** Ms Xuelin Zhang, a PhD candidate sponsored by the Central Research Fund of the Hong Kong Polytechnic University. The funder has played no role in the design of the study, the collection and management of the data, the analysis and interpretation of the data, the writing of the manuscript, and its submission. https://www.polyu.edu.hk/gs/.

**Competing interests:** The authors have declared that no competing interests exist.

deliverable approach could be incorporated into clinical policies and guidelines to support families with CHF without geographic and time constraints.

## Trial registration

ClinicalTrials.gov Identifier: NCT04917159. Registered on 08 June 2021.

## Introduction

Chronic Heart Failure (CHF) is a cardiovascular condition characterized by progressively debilitating symptoms and frequent life-threatening exacerbations [1, 2]. It affects 26 million individuals and their families globally. Patients with CHF frequently suffer increasing exacerbations of multiple debilitating physical signs and symptoms, such as fatigue, swelling, breathlessness, fluid retention, and palpitations, leading to impaired daily physical functioning [3, 4]. Nearly 75% of patients with CHF cannot function independently [5], compelling them to rely on their family caregivers (FCs) to support daily living activities, provide psychological support, and provide assistance in maintaining or improving CHF self-care practices [6].

Quality of life (QOL) is a multidimensional construct that reflects an individual's perception of their physical and mental health [7]. Patients with CHF have generally lower QOL than healthy individuals and those with other chronic conditions [8–10]. The reduced QOL of CHF patients increases their risk of rehospitalization and mortality [11]. It is important to note that the FCs of CHF patients also experience impaired QOL to a degree comparable to that of the patients themselves [12]. A decline in the health status of FCs contributes to an increased risk of developing morbidities such as hypertension and cardiovascular disease [13], and may even lead to premature mortality [14, 15]. Moreover, the diminished health status of FCs is associated with poorer health outcomes for CHF patients due to the compromised quality of the care provided by the FCs [16, 17].

Effectively managing CHF involves essential self-care practices such as medication adherence, limitations on salt and fluid intake, symptom monitoring, regular check-ups, and regular physical activity. These strategies are crucial for enhancing QOL and reducing the risk of readmission due to decompensation episodes [18–20]. However, engaging in CHF self-care activities, such as taking note of food items or seeking professional help, can have the effect of causing the individual to establish a psychological connection to the tangible and potentially distressing consequences associated with CHF, thereby evoking thoughts about the illness and eliciting reactions to its potential dangers. For instance, CHF patients may relate their interactions with healthcare providers to their "sick identity". This may lead them to choose to avoid seeking professional help as a way to deny their CHF disease and cope with feelings of abnormality [21]. FCs are often dedicated to providing competent and timely care [22]. Many FCs unconsciously respond with excessive vigilance in ensuring the well-being and comfort of their patients, driven by their fear of potential future losses [23, 24]. This heightened vigilance may persist even when the patient's health is relatively stable and care demands are low [25, 26].

Studies have shown a high prevalence of anxiety (patients: 55.5%; FCs: 50%) and depression (patients: 22–42%; FCs: 23–47%) among CHF patients and their FCs [27–30]. The majority of FCs also experience elevated levels of perceived caregiving burden [31]. To avoid negative experiences, both patients and their FCs may socially isolate themselves [32, 33] and be reluctant to seek support [25]. Consequently, they may experience tensions in the dyadic relationship [34, 35]. Most patients were unable to adhere to recommended HF self-care behaviors

[36]. These emotional and behavioral responses affect the other person [37], resulting in a strained dyadic relationship between patients and their FCs [38], poorer physical functioning of patients [39], and reduced QOL of both patients and their FCs [35, 40]. Thus, interventions aimed at alleviating the impacts of CHF on the health and well-being of both patients and their FCs are necessary.

Non-pharmacological interventions that primarily target individuals, whether those with CHF or their FCs exclusively, have shown limited success in improving health outcomes [41–44]. Within the broader context of chronic illness, research has shown that interventions targeting both patients and FCs together, known as dyad interventions, are more effective for both patients and their FCs, compared to interventions focused solely on patients or FCs [45, 46]. Positive effects on various health outcomes include physical symptoms, mental health, psychosocial functioning, dyadic relationship, and QOL for both patients and their FCs. Yet only a limited number of experimental studies on CHF have focused on optimizing QOL and related outcomes for CHF patients and their FCs [47]. Therefore, the impact of these interventions on QOL and other psychological outcomes remains inconclusive [48–50]. The commonly used approaches were CBT [48–50] with the aim of identifying and correcting thoughts related to CHF self-care through cognitive efforts for both patients and their FCs. While CBT is a widely used psychological intervention for addressing psychological issues related to chronic physical conditions, it has been criticized for its limited efficacy and low reproducibility [51]. Merely attempting to modify difficult thoughts may have a limited effect on bringing about long-term beneficial changes in the psychological state of a distressed individual. This limitation arises because the efforts of individuals to alter their thoughts may not fully address the broader social and material influences that shape their experiences [52].

Acceptance and Commitment Therapy (ACT), a transdiagnostic therapy grounded in relational frame theory and functional contextualism [53], appears to be well-suited for patients with CHF and their FCs in the context of home care. The aim of ACT is to improve daily functioning and QOL by cultivating psychological flexibility [54, 55]. Psychological flexibility refers to the capacity to recognize situational demands at the present moment and to engage in value-driven actions despite external and internal barriers [53]. ACT emphasizes the psychological, situational, and social contexts that modulate the behavioral influence of thoughts and emotions [53]. Instead of attempting to alter cognitive content, the aim of ACT is to change the function of problematic behavior to allow individuals to behave in line with their values [53]. Through ACT, individuals are guided to accept and embrace their feelings, enabling them to detach themselves from the specific content of their thoughts by fostering mindful awareness of the thinking process, and encouraging them to align their actions with their personal values [53]. ACT has demonstrated effectiveness in reducing anxiety and depressive symptoms and in improving the self-care and QOL of patients with cardiovascular diseases (CVD) [56] and FCs [57]. However, it is worth noting that a significant percentage of existing studies on ACT primarily focus exclusively on either patients or FCs.

To date, only two studies have adopted an ACT intervention for dyads, and in both cases the interventions were delivered via telephone in an individual/dyad format for specific cancer conditions, such as lung cancer [58] or gastrointestinal cancer [59]. Both studies involved relatively small sample sizes ranging from 40 to 50 dyads. Although no statistically significant differences were found in QOL, use of health services, and caregiver distress after the intervention when compared to those who received the usual care, high completion rates and levels of satisfaction among the participants in these studies provide early support for the use of a remote ACT intervention for patient-FC dyads.

Remote interventions, delivered through telephone and video, have emerged as promising alternatives to address the challenges of the scalability and accessibility of patient-FC dyad

interventions [47]. Compared to in-person interventions, remote approaches have shown similar effects on improving psychological outcomes [60]. Notably, videos can provide important visual information that telephone interventions cannot, allowing for the observation of non-verbal cues that may be relevant to psychological status and enhancing the therapeutic relationship between participants and interventionists [60]. In addition, while the effects of ACT delivered in a group format have been found to be equivalent to individual-based interventions [61], the group format offers unique therapeutic benefits. These benefits include reducing feelings of isolation, facilitating shared learning from others' experiences, and providing opportunities for modeling new coping strategies and behaviors [62]. Therefore, a videoconferencing group format was chosen as the mode of delivery for this intervention.

## Methods

### Objectives

This study is proposed with the objective of evaluating the effectiveness of a videoconferencing group-based dyad ACT intervention for CHF patients and their FCs compared with CHF education only at a three-month follow-up on: (1) Patient and FC outcomes: QOL (primary outcome), anxiety, depressive symptoms; perceived quality of the relationship, psychological flexibility, and self-compassion; (2) Patient-only outcomes: Healthcare service utilization and CHF self-care behavior; and (3) Caregiver-only outcome: Caregiving burden.

### Study design

The proposed study is a two-armed, parallel-group, equivalence randomized controlled trial (RCT). Dyads of CHF patients and their FCs will be randomized at a 1:1 ratio to either the intervention group, which will receive an ACT-based intervention, or the control group, which will receive only CHF education.

### Study setting

The participants will be recruited from the Department of Cardiology at Taihe Hospital in Shiyan City, Hubei province, China. The hospital is the largest public hospital in Shiyan City, which has a total population of 3,209,004, according to the 2020 census [63]. The Department of Cardiology is responsible for the prevention, treatment, and rehabilitation of residents with cardiovascular disease, including CHF. Annually, about 4,600 patients are hospitalized with a diagnosis of CHF. After CHF patients are discharged from the hospital, they will be expected to visit the heart failure clinic to review their health condition every one to three months. They will also receive the usual care, which is in line with national and local clinical CHF guidelines in mainland China [64]. The usual care includes follow-up telephone calls to provide educational support based on the inquiries of patients and to remind patients of the necessity of undergoing a medical check-up conducted by a physician one week after discharge and by a clinical nurse one month after discharge.

### Eligibility

Participants of the present study will be both patients with CHF and their corresponding primary FCs. CHF patients will be eligible if they are 18 years of age or older, have been clinically diagnosed with CHF [65] as indicated in their electronic medical records, and with a New York Heart Association (NYHA) Classification of I to III as confirmed by a physician in this study, have been hospitalized in the past one year, and are residing in a household with family members. Patients will be excluded from the program if they have received a score of 0–2 on

the Mini-Cognitive scale [66], indicating cognitive impairment, or have a documented medical history of psychiatric illness, dementia, or a life-threatening illness, such as severe pulmonary disease or end-stage renal failure or are living in a nursing home. CHF patients will be asked to nominate their primary FC, who would be the person with the highest average number of contact hours among their family members, to join the program. FCs should be 18 years of age or older, and can be the spouse, parent, or adult child of the patient. FCs will be excluded if they are paid caregivers, have a medical history of psychiatric illness, dementia, or a life-threatening illness such as severe pulmonary disease or end-stage renal failure, or are providing care to multiple patients in the family.

In addition, at least one member of the patient-FC dyad will be required to have a smartphone installed with the WeChat app and a data plan. Both members should be able to understand Chinese and communicate in the language, as well as be able to access Tencent VooV Meeting via smartphone to attend the sessions.

Tencent's VooV meeting is the most common streaming platform for videoconferencing in China [67]. It offers multiple functions, such as real-time screen sharing, online document collaboration, small group discussions, and text messaging to improve experiential learning, interactivity, and cooperation during meetings [68]. The VooV meeting includes a mini program embedded through WeChat. WeChat, which has been installed in 90% of smartphones in China, is the country's most popular social media application [69]. It features various platforms, such as text or voice messaging, real-time video or voice communications, and payment, and has been integrated into every aspect of human life in China [70].

## Sample size calculation

Power calculations necessitate specific information on health outcome indicators from previous studies [71]. However, due to the unavailability of detailed dyadic information specifically for dyadic analysis on QOL and other health outcomes for CHF patients and their FCs, the prior power analysis conducted using G*Power in this study relied on individual-level data. Previous studies in CHF have shown that mean differences in QOL between patients who received a psychosocial intervention and the usual care ranged from 0.11 to 0.68 [72]. For ACT interventions targeting QOL, a recent review of meta-analyses reported a mean effect of 0.48 for the clinical and non-clinical population, ranging from 0.37 to 1.55 when those who received the ACT intervention were compared to the active control, not including CBT [73]. Thus, an estimated effect size of 0.50 was adopted across the above reviews. Targeting that effect size at a power of 0.8 with a prior power calculation, a minimal sample size of 128 will be required to detect the difference between the ACT group and control group. Considering a weighted dropout rate of 16% for those participating in an ACT intervention indicated in a recent meta-analysis [74], and given group-setting and patient-FC dyadic participant requirements for the interventions, the total sample size was increased to 160 patient-FC dyads, accounting for an anticipated dropout rate of 20%.

## Randomization and blinding

In this study, allocations with a ratio of 1:1 will be performed through an online randomization program (https://www.randomizer.org/) by an independent researcher, who will be excluded from any other activity throughout the intervention. The unit of randomization will be the patient-FC dyad. The randomization list will be stratified by the type of relationship between the patients and caregivers (e.g., spousal relationship vs. non-spousal relationship) with a block size of four. The allocation sequence will be printed and sealed separately in opaque envelopes by the independent researcher.

The staff nurses in the study hospital will enroll dyads of patients and caregivers. After the baseline assessment is completed, the registration data of each dyad will be given to the research coordinator. The research coordinator will open the envelope and then enter the data into the database and assign dyads to interventions. The participants will be informed of their allocation status via smartphone when the research coordinator plans to schedule the first session.

The research assistants serving as outcome assessors in this study will be blinded to the group allocations; they will be unable to access the participants' intervention conditions during the program. In addition, outcome assessors will remind the participants to remain blind to their group assignment during the outcome assessments. If this blinding is breached for any reason, another research assistant who is unaware of the group condition of the dyads will take over the phonic assessment. Participating dyads and interventionists will not be blinded because of the nature of the psychoeducational intervention.

## Interventions

Participating dyads will be allocated to either the intervention group or the control group. All of them will receive four identical weekly sessions via the Tencent VooV meeting over four consecutive weeks in addition to the usual care provided by the Department of Cardiology. Each session will include four to eight dyads and last approximately two hours. Participants will receive relevant materials and handouts prior to the first session, and will be instructed to retain these materials and handouts throughout the duration of the study and beyond. The online sessions will occur at the same time slot for both study groups and be scheduled during weekday evenings and weekend afternoons to accommodate the typical family events and work schedules of the participants. During the period of the interventions, WeChat voice calls or text message reminders will be sent to the dyads before each session to improve participant retention and support engagement in the program.

Patients and their FCs will be required to participate in the group sessions on a dyad basis. During the intervention period, if either member of the dyad has been hospitalized, both will be asked to suspend their participation in the program. Hospitalization typically requires individuals to prioritize their acute medical needs. After being discharged from the hospital and returning home, the patient-caregiver dyads will be asked to resume their participation within one week after discharge. If patient-FC dyads miss a session, a make-up session will be provided via videoconferencing within three days before the start of the next scheduled session.

## Intervention group: ACT-based intervention (ACT group)

Each session in the intervention group will consist of group ACT (1.5 hours) and brief CHF education (0.5 hours) conducted by two trained registered nurses as facilitators. The ACT intervention covers six interrelated core processes to produce psychological flexibility by using metaphor illustrations, experiential exercises, and guided mindfulness exercises, while the contents of the brief CHF education only include information on CHF self-care. Prior to the session, the participant will receive an additional packet supporting the practice of ACT exercises in the sessions, which will include black and white cards, whiteboard markers, two small packages of raisins, and one towel, in addition to one set of CHF self-care materials. The handouts for this group will consist of information on CHF self-care, ACT skills, and homework assignments making changes to the ACT processes targeted in each session.

*Development of the ACT-based intervention*. The first version of the ACT intervention protocol was informed by the ACT classic therapist manual [53, 75–77] and by literature on the experiences of CHF patients and FCs [78, 79], previous ACT trials with CVD [56] and FCs

[57], and an ACT intervention at the dyadic level [58, 59]. The intervention protocol (Version 1) was initially evaluated in a four-week uncontrolled clinical trial using an in-person format (*N* = 7). Several adaptations were made, such as emphasizing the interdependent nature of the dyadic relationship, adopting ACT experiential exercises to accommodate patient-FC dyads rather than individual patients or caregivers, and utilizing in-session dyadic techniques such as role-play and mutual elicitation of feedback. Afterward, the intervention protocol was refined and adopted in an online videoconference format (Version 2). The intervention protocol (Version 2) was retested in a four-week pilot RCT via smartphone without a follow-up assessment (*N* = 16); the findings indicated that further modifications were required on the homework assignments and facilitating instructions on some experiential exercises. Revisions were accordingly made to the intervention protocol (Version 3) before being implemented in this RCT. All versions were finalized by the research team and informed by feedback from the participants and facilitators, direct observations by the facilitators during the delivery of the intervention, and input from the ACT expert. Any disagreements were discussed with an experienced ACT researcher (YWM) and resolved by consensus.

A four-session group-based ACT intervention has been deemed adequate for patients and their FCs based on previous evidence from systematic reviews [54, 57]. For patients with chronic diseases, a systematic review found that ACT interventions consisting of less than five group sessions led to significant medium-to-large improvements in disease management compared to the usual care [54]. These improvements were observed in various health indicators, such as QOL, disease self-care behavior, and physical functioning, in patients with epilepsy, cancer, diabetes, and cardiac conditions [54]. As for the FCs of patients with long-term illnesses, they showed significant improvement in various health outcomes after two to four group sessions of an ACT intervention when compared with the usual care, as indicated in a recent systematic review [57]. These outcomes included: anxiety, depression, QOL, and family functioning [57].

The four-session group-based ACT intervention in this study includes exploring the control agendas of the participants, identifying their values, exploring their thoughts and feelings, and finding ways for them to take committed actions in line with the personal values that they hold. The first ACT session will explore their control-based coping strategies in an attempt to help the participants cope with their private experiences and the workability of control when pursuing a meaningful life. The following sessions will start with a review of the psychological processes in which the participants were instructed in the previous session and the between-session assignments. From the second session, the facilitators will focus on increasing the participants' connection with their life values and their commitment to values-based actions. Individual values will be identified, while alternative behavior related to the participants' values will be clarified. Patient-FC dyads will have the opportunity to share their values and alternative value-based actions to manage the CHF situation with each other. Potential cognitive and emotional barriers in the context of home care to adopting alternative behavior and preserving values-driven long-term action will be explored and addressed via metaphor illustrations and experiential exercise practice. Throughout the program, participants will be encouraged to be more accepting and compassionate, detach themselves from self-conceptualizing, take a flexible perspective, and develop flexible attention. In addition, dyadic sharing and reflecting on their experiences will be highlighted, which in turn will promote the establishment of supportive relationship bonds.

*Facilitator training and fidelity checks.* Due to the experiential learning and load of the online course material, the ACT-based intervention will be delivered to the patient-family caregiver dyads by two registered nurses working together. The primary facilitator (XZ) is a doctoral student in nursing who has completed a total of 37 days of ACT workshops led by

ACT experts worldwide and in China. The co-facilitator (Ms. Chunmei Xiao) is a registered nurse at a local hospital who holds a bachelor's degree in nursing and has received two days of introductory-level training in ACT led by ACT experts in China. This training equipped her with the foundational knowledge and skills necessary to assist in delivering the online ACT intervention.

Both facilitators have a minimum of three years of experience working with cardiac inpatients and have prior experience in co-facilitating group ACT-based interventions for patient-caregiver dyads during the pilot studies, where they together conducted approximately 8 group sessions for CHF conditions (a total of 12 hours). All of the sessions in these pilot studies were video recorded for quality assurance and fidelity checking. These recordings were then reviewed and discussed with the experienced ACT researcher (YWM). To further enhance their expertise in delivering the ACT-based intervention for the current study, prior to commencing the study the facilitators have practiced role-playing under supervision using the finalized version of the intervention protocol for this study.

To monitor the competence of the facilitators and the fidelity to the contents of the intervention protocol, each group session will be video recorded with the consent of the participants. The recordings will be reviewed independently by the facilitators immediately after the session, by using the ACT core competency self-rating form [80]. Feedback on intervention fidelity of both facilitators will be discussed during in-person meetings after each session. In addition, regular weekly supervision and timely supervision will be provided by the experienced ACT researcher (YWM) to ensure adherence to the intervention protocol throughout the program. Any difficulties encountered during the sessions will be reviewed and discussed. If any areas for improvement are identified, the primary facilitator will provide a remedy before the next session.

## Control group: CHF education

CHF education only as an active control was chosen because knowledge of CHF is a foundational and essential component to empower patients with CHF and their FCs to manage CHF in the family context [81]. The sessions for the CHF education control will be delivered by one registered nurse (Ms. Qiaoyun Jin) who holds a master's degree in nursing and has at least three years of experience working with cardiac inpatients. Each two-hour session will include a review of the previous session, didactic education, and a Q&A section to evaluate the participants' understanding of the key concepts based on established CHF knowledge scales, including the Heart Failure Knowledge Test [82] and the Dutch Heart Failure Knowledge Scale [83]. Apart from information on CHF self-care, the control group will receive additional knowledge on CHF, involving the definition of CHF, epidemiology, diagnosis, comorbidity, and treatment. The contents of the CHF education have been mapped based on the latest national clinical practice guideline for CHF [64].

Prior to the sessions, dyads will be provided with one set of CHF self-care materials, including two salt spoons, one portable pill box, a scale and cup measurement, and printed handouts. The handouts for the control group will consist of information on CHF self-care. Table 1 lists a summary of the sessions for the ACT-based intervention group and Control group of CHF patients and their FCs.

## Outcomes

The characteristics of the participants will be gathered at baseline before the intervention, including: (1) their sociodemographic data, such as their age, gender, educational attainment, employment, and relationship with each other; and (2) their clinical data, such as the patients'

**Table 1. Summary of the ACT group and control group sessions.**

| Session | ACT group (Intervention group) | CHF education group (Control group) |
|---|---|---|
| Week 1 Session 1 | Introduction to the sessions, guidelines, and group expectations<br>Introducing and practicing mindfulness when eating raisins<br>Exploring the control agenda: identifying issues, the normalcy of psychological suffering, and the unworkability of the control-based strategies by practicing guided mindfulness to revisit a challenging moment during the CHF management process in the family context<br>Introducing the paradoxical effects of control-based strategies and acceptance as an alternative by using the metaphor of quicksand<br>Providing an overview of CHF self-care and symptom monitoring<br>Homework assignment: to identify challenging moments and be aware of inner events | Introduction to the sessions<br>Providing information on CHF facts and self-care<br>Definition of heart failure<br>Terminology of heart failure<br>Heart failure epidemiology and prognosis<br>Key steps in the diagnosis of CHF<br>Importance of CHF self-care*<br>Importance and strategies of monitoring symptoms*<br>Reviewing the main concept taught in this session based on the Heart Failure Knowledge Test [82] |
| Week 2 Session 2 | Practicing mindfulness when listening<br>Reviewing the previous session<br>Clarifying personal values with a "value in trash" exercise and exploring how individuals may choose to respond to negative thoughts/feelings about managing CHF in a value-consistent manner<br>Exploring the unworkability of attempts to avoid or control negative internal experiences [e.g., relationships, thoughts, feelings) and how these efforts lead to value-inconsistent actions and impaired QOL by illustrating the metaphor of "the passengers on the bus"<br>Providing information on recognizing the most common cardiac medication regime, the correct use and possible side effects, and the importance of medication adherence and possible solutions<br>Homework assignment: identify an alternative behavior they are willing to perform that would help them move toward their core values | Reviewing the previous session<br>Providing information on CHF pharmacological treatment and medication adherence<br>Pharmacological treatment for CHF<br>Correct use and possible side effects*<br>Importance of medication adherence and possible solutions*<br>Reviewing the main concept taught in this session based on the Dutch Heart Failure Knowledge Scale [83] |
| Week 3 Session 3 | Practicing mindfulness when breathing<br>Reviewing the previous session<br>Identifying an alternative behavior they are willing to perform that would help them move towards their core values<br>Exploring possible inner barriers (e.g., cognitive or emotional) to taking these committed actions<br>Promoting detachment from unhelpful thoughts, feelings, and prior relationship experience about CHF and CHF management in the context of home care by engaging in defusion exercises<br>Providing information on the importance of fluids and dietary management, a low-sodium diet, and tips on selecting food every day<br>Homework assignment: establish an action plan for an identified alternative behavior, implement the specific behavior, and be mindful of the response of family members. | Reviewing the previous session<br>Providing information on CHF treatment, dietary and fluid management<br>Cardiovascular and non-cardiovascular comorbidities<br>Oxygen therapy and ventilatory therapy<br>Mechanical circulatory support<br>Heart transplantation and renal replacement<br>Importance of fluids and dietary management*<br>Low-sodium diet*<br>Tips on selecting food every day*<br>Reviewing the main concept taught in this session based on the Heart Failure Knowledge Test [82] and the Dutch Heart Failure Knowledge Scale [83] |

(*Continued*)

**Table 1.** (Continued)

| Session | ACT group (Intervention group) | CHF education group (Control group) |
|---|---|---|
| Week 4 Session 4 | Practicing mindfulness in a brief body scan exercise<br>Reviewing the previous session<br>Clarifying helpful steps to take for patients and their FCs when painful experiences are aroused<br>Allowing participants to extend self-understanding and self-compassion to their family members by practicing guided mindfulness and revisiting their past experiences of conflict in the context of family<br>Summarizing the ACT skills taught in the session<br>Identifying some possible future challenges and knowing how these skills may be useful<br>Providing information on CHF physical activities, and cigarette and alcohol consumption | Reviewing the topics discussed in previous sessions<br>Providing information on CHF physical activities, and cigarette and alcohol consumption<br>Advantages and disadvantages of regular physical exercise*<br>Common principles during exercise*<br>Monitoring pulse rate to adjust the intensity of exercise*<br>Recognizing and responding to possible cardiac symptoms during the exercises*<br>The benefits of smoking cessation and common cessation methods *<br>Abstaining from excessive alcohol intake*<br>Reviewing the main concepts taught in this session, based on based on the Heart Failure Knowledge Test [82] and the Dutch Heart Failure Knowledge Scale [83] |

Note: ACT: Acceptance and commitment therapy; CHF: Chronic heart failure; QOL: Quality of life. * The content is similar to that in the corresponding session for the intervention group.

NHYA classification, length of time that they have had the disease, their comorbidities, and their caregivers' length of caregiving.

The effectiveness of the interventions will be evaluated at baseline, immediately post-intervention, and at three months post-intervention. The research assistants have been fully trained in outcome assessments to ensure quality and consistency. They will collect data through telephone interviews of patients with CHF and their FCs.

## Primary outcomes

The primary outcomes are the patients' QOL and that of their FCs. Generic measures, such as the EuroQol five-dimensional five-level (EQ-5D-5L) scale and the EuroQol visual analog scale (EQ-VAS) [84], will provide a broad evaluation of QOL across different populations and medical conditions. Disease-specific measures, such as the short form of the Kansas City Cardiomyopathy Questionnaire (KCCQ) [85], can be used to capture CHF-related issues that are particularly important to patients with CHF, and that are often sensitive to change [86]. The evaluation of the patients' QOL will be conducted using different scales: the KCCQ [85], EQ-5D-5L, and EQ-VAS [84]. These measures provide a comprehensive assessment of patient QOL from different perspectives [86]. As for the FCs, their QOL will be assessed using the EQ-5D-5L and EQ-VAS [84].

*Disease-specific QOL.* Disease-specific QOL will be measured by employing KCCQ, which consists of 12 items measured on a five or seven-point Likert scale with an overall score of between 0 to 100, with higher scores reflecting better QOL [85]. This scale has demonstrated high responsiveness, test-retest reliability, and prognostic ability [85].

*Generic QOL.* The EQ-5D-5L and EQ-VAS will be used to evaluate generic QOL. The EQ-5D-5L is comprised of five items with five levels. The five items can result in a five-digit number, which can be adapted to a single utility score by utilizing the Chinese scoring algorithm [87]. The EQ-5D-5L has demonstrated good reliability with a Cronbach's α of 0.857, and validity in the Chinese population [88]. The EQ-VAS is comprised of a single global rating. Potential scores range from 0 to 100, with a higher score reflecting a better perceived health status

[87]. The reliability of the scale has been demonstrated with a Cronbach's $\alpha$ of 0.83 among Chinese [89].

## Secondary outcomes

The secondary outcomes include measures at the dyadic and individual levels for patients and their caregivers.

For patients and their family caregivers:

Severity of anxiety

The Generalized Anxiety Disorder Scale-7 (GAD-7) will be utilized to test the level of severity of an individual's anxiety. The GAD-7 consists of 74-point Likert items. The total score ranges from 0 to 21, with lower scores representing less severe anxiety [90]. The GAD-7 has shown good reliability, with a Cronbach's $\alpha$ of 0.91, and construct validity among Chinese [91].

Severity of depressive symptoms

The Patient Health Questionnaire (PHQ-9) will be employed to evaluate the level of severity of an individual's depressive symptoms. The PHQ-9 covers nine items measured on a four-point Likert scale. The total score ranges from 0 to 27, with lower scores representing less severe depression [92]. The Cronbach's $\alpha$ is 0.91, with good sensitivity and specificity among Chinese [91].

Perceived quality of the relationship

The short form of the Dyadic Adjustment Scale (DAS-7) will be adopted to examine the perceived quality of the relationship between the patients and their caregivers. The DAS-7 consists of seven items measured on a six or seven-point Likert scale. The total score ranges from 0 to 36, with a higher score indicating a higher-quality relationship. It was used among CHF patients and caregivers and found to have acceptable Cronbach's $\alpha$ coefficients of 0.70–0.78 [93].

Self-compassion

The short form of the Self-compassion scale (SCS–SF), which is comprised of 12 items, will be utilized to measure self-compassion [94]. The Chinese version of the SCS–SF demonstrated acceptable reliability, with a Cronbach's $\alpha$ of 0.686, and validity among Chinese [95].

Psychological flexibility

The Comprehensive Assessment of Acceptance and Commitment Therapy Processes (CompACT) [96] will be utilized to evaluate the psychological flexibility of individuals. The Chinese version of CompACT consists of 18 items measured on a seven-point Likert scale. Scores range from 0 to 126, with a lower score representing a greater level of psychological flexibility. With a Cronbach's $\alpha$ of 0.87, the validity and reliability of CompACT have been demonstrated in a non-clinical sample in China [97].

Patient-only outcomes:

Healthcare service utilization

The frequency of all course and CHF-related hospitalizations and emergency department visits in any hospital will be assessed by self-reports.

CHF self-care behavior

The CHF self-care behavior of patients will be evaluated using the European Heart Failure Self-care Behavior Scale (EHFScBs). The ECHFScBs is comprised of 12 items measured on a five-point Likert scale. It is used to evaluate CHF self-care behavior, with an emphasis on help-seeking and regimen-complying behavior [98]. The possible total score ranges from 12 to 60, with a higher score indicating a lower level of self-care performance. The Chinese version of the ECHFScBs has demonstrated good reliability, with a Cronbach's $\alpha$ of 0.82, and validity among Chinese patients with CHF [99].

Caregiver-only outcome:

Perceived caregiving burden

The Zarit Caregiver Burden Interview (ZBI) will be used to evaluate feelings of caregiving burden [100]. It consists of 22 statements measured on a five-point Likert scale, with a total score ranging from 0 to 88. The Chinese version has shown good internal consistency (Cronbach's α 0.875) among Chinese caregivers and good validity [101].

## Procedure

Potential CHF patients will be initially identified by the staff nurses through the reviewing of electronic medical records. The staff nurses will contact the designated contact persons of the patients via telephone to obtain the contact information of the patients and their FCs. Then, both the patients and their FCs will be approached by phone and will be provided with an introduction to the purpose and procedure of the study. If one member of a patient-FC dyad is unable to participate, they should inform the research team. If there is no response, the staff nurse will follow up with a phone call to provide further information on the study if needed. Interested patient-FC dyads will be invited by the staff nurse to attend either through an online assessment via WeChat or an in-person clinical visit to confirm eligibility.

After the eligibility of the patient and the patient's FC is confirmed, the staff nurses at the Department of Cardiology will provide a detailed explanation of the study's aim, methods, potential benefits, and risks, including all necessary elements of informed consent. They will do this either during the patient's regular visit to the clinic or online via a WeChat video call. Once oral consent is given, the patient-FC dyads will receive a link to the information sheet via the WeChat platform. Verbal recorded informed consent will be sought from each CHF patient and their FC within one week after the documents have been sent out over the smartphone. A detailed schedule and timing of the evaluations are presented below in Fig 1. T5 and T6 represent the time immediately post-intervention and at the three-month follow-up point, respectively. The flow of the study is presented in Fig 2.

Technical support will be provided before and during the sessions. Prior to the sessions, a video with step-by-step instructions on how to log onto VooV Meeting via a WeChat mini program will be provided to each dyad. Research assistants will conduct brief tutorials with the dyads, either in person in the clinic or online via smartphone, in the lead up to the first session to help them set up VooV meetings via the WeChat mini program. This is to ensure that they are able to access the group sessions and check their audio and video connections during the sessions. Throughout the program, research assistants will also provide timely online support on technical issues for the patient-FC dyads.

To promote recruitment and retention, the dyads will be given monetary incentives to undergo each assessment during the study, as compensation for their time, via the WeChat platform. For example, they will be given 50 Yuan RMB (≈6.97 USD) for each session and each assessment at baseline and post-intervention, and 100 Yuan RMB (≈ 13.95 USD) for the three-month follow-up assessment.

## Data management

A detailed database will be established to record the assessments and progress of each participant. To preserve confidentiality, each participant will be assigned a unique identifier. This database will be password-protected and only accessible to authorized members of the research team. The researcher coordinator will periodically monitor the data collection process and the safety of the intervention. Any adverse events will be identified, documented, and handled with caution. Should any disagreements arise, the research team, which will include

| | Study period | | | | | | | | |
|---|---|---|---|---|---|---|---|---|---|
| | Enrolment | Baseline | Intervention (weekly) | | | | Post-intervention | 3-month follow-up | |
| **Timepoint** | -t1 | t0 | t1 | t2 | t3 | t4 | t5 | t6 | |
| **Enrollment** | | | | | | | | | |
| Eligibility screening | PT/FC | | | | | | | | |
| Informed consent | PT/FC | | | | | | | | |
| Allocation | | PT/FC | | | | | | | |
| **Intervention implementation** | | | | | | | | | |
| Intervention group (ACT-based intervention) | | | ←→ | | | | | | |
| Control group (CHF education only sessions) | | | ←→ | | | | | | |
| **Assessments** | | | | | | | | | |
| Participant sociodemographic and clinical data | | PT/FC | | | | | | | |
| Five-dimensional five-level EuroQol (EQ-5D-5L) | | PT/FC | | | | | PT/FC | PT/FC | |
| Visual analog scale (VAS) in EQ-5D-5L | | PT/FC | | | | | PT/FC | PT/FC | |
| Generalized Anxiety Disorder Scale-7 (GAD-7) | | PT/FC | | | | | PT/FC | PT/FC | |
| Patient Health Questionnaire-9 (PHQ-9) | | PT/FC | | | | | PT/FC | PT/FC | |
| Dyadic Adjustment Scale-7 (DAS-7) | | PT/FC | | | | | PT/FC | PT/FC | |
| Comprehensive Assessment of ACT Processes (CompACT) | | PT/FC | | | | | PT/FC | PT/FC | |
| Self-compassion scale (SCS) | | PT/FC | | | | | PT/FC | PT/FC | |
| Perspective-taking subscale in the Interpersonal Reactivity Index (IRI) | | PT/FC | | | | | PT/FC | PT/FC | |
| Self-reported healthcare service utilization | | PT | | | | | | PT | |
| Short form of the Kansas City Cardiomyopathy Questionnaire (KCCQ) | | PT | | | | | PT | PT | |
| European Heart Failure Self-care Behavior Scale (EHFScBS) | | PT | | | | | PT | PT | |
| Zarit Caregiver Burden Interview (ZBI) | | FC | | | | | FC | FC | |

**Fig 1. Schedule of enrollment, interventions, and assessments for each dyad of the study.** Note: ACT: Acceptance and commitment therapy; FC: Family caregiver; PT: Patient.

cardiologists, nurses, and academics, will engage in discussions and reach a consensus. Consultations with clinical trial specialists in the study hospital will be sought if required.

## Statistical analysis

All quantitative data collected from the participants will be coded numerically and analyzed using IBM SPSS 27. Baseline sociodemographic data, clinical data, and outcome variables will be reported using medians, proportions, and counts for categorical data; and means, standard deviations (SDs), skewness, and kurtosis for continuous variables for the participants in the two groups. Differences in the participants' characteristics at baseline will be explored using an independent T-test or Mann-Whitney U test for continuous variables, and a Chi-squared or Fisher's Exact test for categorical variables. Data will be analyzed following the intention-to-treat principle. To address missing data, multiple imputations will be employed with 50 imputed samples [102]. For the outcomes variables reported by patients and FCs, multilevel modeling (MLM) is recommended to account for the nonindependence of data within the dyad and the presence of repeated measures [103]. In cases where health outcomes apply exclusively to either the CHF patient or their FC, the analysis will employ individual MLM, which will involve interaction effects and the main effects of the study group and time. Assessment time points (baseline, immediately after the intervention, and three months post-

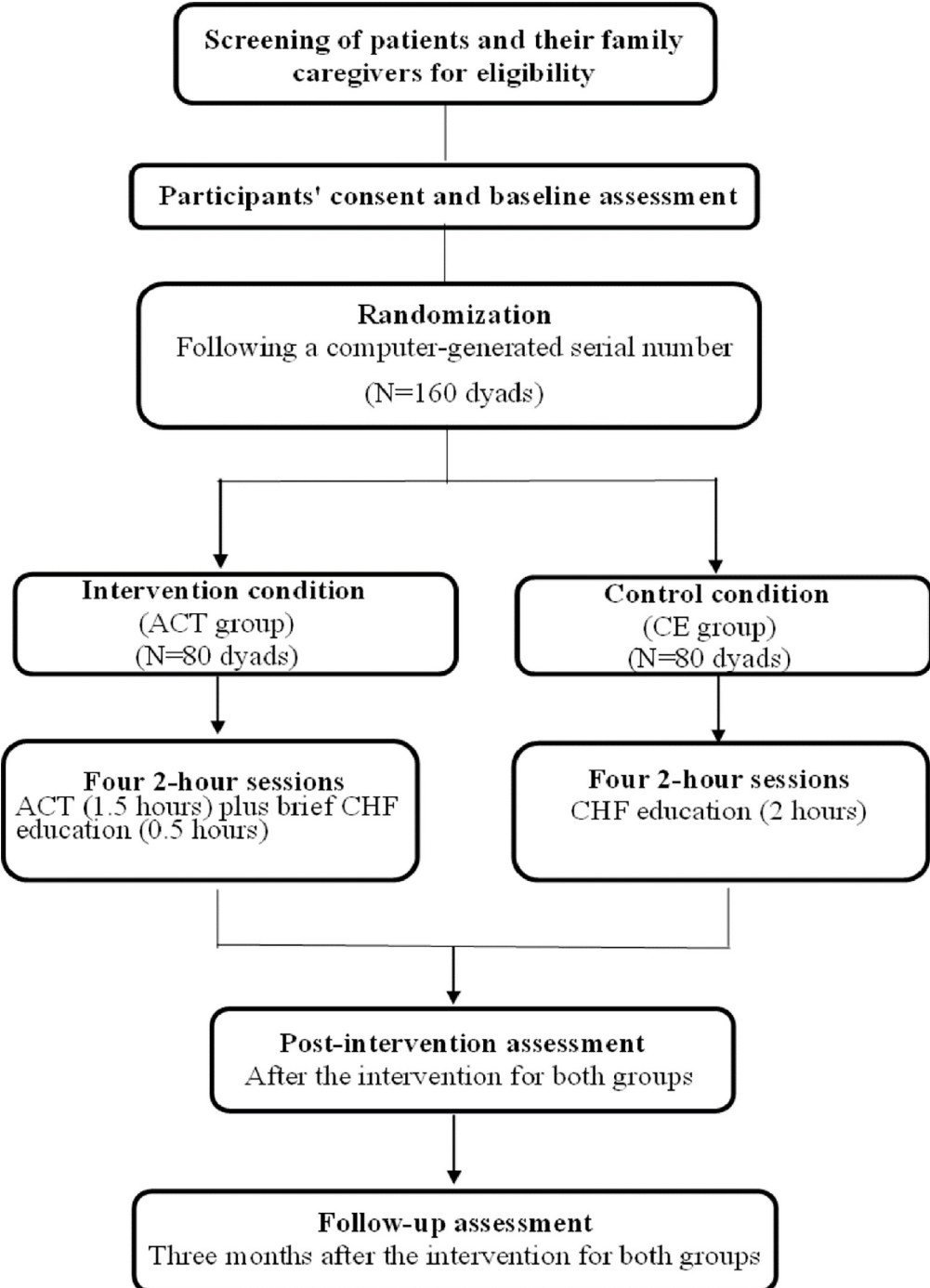

**Fig 2. Design of the present study.** Note: ACT: Acceptance and commitment therapy; CHF: Chronic heart failure; CE: CHF education.

intervention) are treated as categorical variables. Response variables that follow a normal distribution will be tested using the multilevel normal model with a linear link function. With response variables that display skewness in their distribution, the multilevel Poisson model with a log link function will be employed. To healthcare services utilization measures (e.g., the

number of hospitalizations and emergency department visits), which are ordinal and involve count data, the multilevel Poisson model with a log link function will be applied. For outcomes that apply to both CHF patients and their FCs, MLM for dyadic data will be used [103] In the dyadic models, the fixed-effects parameters encompass the main effects of the study group, time, and role (patient vs. FC), along with their two- and three-way interaction effects. The significance of the group-by-time interaction effect will be used as evidence of the effects of the intervention. The three-way interaction among the study group, time, and role will be examined to assess the differential effects of the intervention on patients and FCs. The random-effects parameters account for separate residual variances for CHF patients and FCs, and the covariance between the residuals reflect the similarity in scores between the two members within the dyad at a specific time point while accounting for the fixed effects. Random intercepts for dyads will be incorporated into the variance of the model in the average outcome across patient-caregiver dyads. A significance level of $p < 0.05$ (two-tailed) will be utilized to determine statistical significance. A partial correlation coefficient ($pr$) will be computed as the effect size measure for each fixed effect [104]. Among participants who complete the questionnaire, Cohen's $d$ will be computed to assess the between-group effect for primary and secondary outcomes. In addition, subgroup analysis will be adopted for specific factors (e.g., intervention completion and non-completion).

## Ethical considerations

Ethical approval for this study was given by the institutional review board (IRB) of the Hong Kong Polytechnic University (reference: HSEARS20210225006; 23-Apr-2021) and the Taihe Hospital (Version 3; reference: 2022KS013; 3-Jun-2022). The study protocol was registered in ClinicalTrials.gov (Identifier: NCT04917159; Registered on 08-Jun-2021). Oral consent and verbal recorded informed consent will be obtained from each patient and FC before the baseline assessment and randomization are conducted. Participation in the study is completely voluntary for CHF patient-FC dyads, and they are free to withdraw from the study at any point without consequences. The participants' identities and data, including information collected and video/audio records, will be kept confidential and anonymous to safeguard their privacy. Only authorized personnel will be allowed access to the data for the purpose of analysis.

## Discussion

This protocol presents an RCT design that compares the effectiveness of an online group-based ACT-based intervention in fostering QOL and related health outcomes in CHF patients and their FCs. The online group-based ACT intervention is expected to improve QOL, psychological symptoms (e.g., anxiety and depression), perceived health status, perceived quality of relationships, self-compassion, and psychological flexibility for patients and FCs, as well as the perceived burden of FCs, CHF self-care behavior, and the healthcare service utilization of CHF patients.

This study has three key strengths. First, it differs from most studies of patients with CHF alone or studies of the FCs of CHF patients alone. The target population is dyads of patients and caregivers with CHF. By expanding the intervention to their FCs, CHF patient-FC dyads can improve their daily functioning and help them to compensate for each other, which may produce synergistic benefits on health at the family level. Second, this study integrated ACT, a transdiagnostic approach, for those with comorbid physical difficulties and emotional and behavioral challenges, which is highly valuable for improving the QOL and other related health outcomes of families with CHF. Third, the online format for most research activities in this study can provide more opportunities for a broad range of families who might otherwise be

unable to participate owing to geographical and time constraints. As subject recruitment will be conducted via smartphone, all eligible CHF patients who have been discharged from the study hospital can be invited to participate in the program with their FCs. This could potentially lower the threshold for participation in the program and may result in the inclusion of a more diverse and extensive range of participants. Low-threshold interventions in this group are needed because impaired QOL is highly prevalent.

However, this study has several limitations. First, due to the online format, the present study may exclude families who are unfamiliar with the Internet and the WeChat platform. CHF patients and their FCs who are older and who have a low level of internet literacy may not be able to take part in this study, which would lower the representativeness of the study sample. Second, involving online interventions might be challenging for participating dyads of CHF patients and their FCs, which may potentially cause attrition. However, an estimated 20% attrition rate was considered when calculating the sample size. In addition, the incentives for participating in a dyad will depend on the number of activities on offer. These are likely to lead to a decrease in attrition for the intervention and study.

The number of CHF patients is growing along with an increasingly aging population in China. Such an increase will impose a heavy burden not only on individuals and families, but also on our society at large. Healthcare costs will proliferate due to an increase in hospital costs to care for these patients and to a reduction in family caregiver productivity. Thus, it is important to have an effective intervention to defray the costs of readmitting patients with CHF to hospital. If this intervention that is delivered online via smartphone proves to be effective, a multi-site RCT can be applied to further promote the effectiveness of the program. In the long run, the intervention could be incorporated into clinical policies and CHF guidelines to support families with CHF without geographical and time constraints. In addition, this family-based intervention could be utilized by families with other chronic diseases to improve their QOL.

## Dissemination

The findings from this study will be published in referred journals. During the enrollment process, participants will be asked if they are interested in receiving the findings or publications emanating from the present study. The results will also be disseminated at national and international academic conferences and research seminars to promote knowledge sharing and guide healthcare initiatives for the broader CHF family.

## Supporting information

**S1 Checklist. SPIRIT checklist.**
(DOCX)

**S1 Protocol. Project submitted to the ethics committee.**
(DOCX)

## Acknowledgments

The authors would also like to thank Ms. Chunmei Xiao and Ms. Qiaoyun Jin for their substantial contributions in sharing their previous challenges in educating chronic heart failure patients and their family caregivers about self-management strategies in chronic heart failure. They will both act as facilitators in either intervention condition. We also thank Dr. Doris YP Leung for her statistical advice.

## Author Contributions

**Conceptualization:** Xuelin Zhang, Grace W. K. Ho.

**Funding acquisition:** Xuelin Zhang.

**Investigation:** Xuelin Zhang.

**Methodology:** Xuelin Zhang, Grace W. K. Ho, Yim Wah Mak.

**Project administration:** Xuelin Zhang.

**Resources:** Xuelin Zhang, Yim Wah Mak.

**Supervision:** Grace W. K. Ho, Yim Wah Mak.

**Validation:** Yim Wah Mak.

**Writing – original draft:** Xuelin Zhang, Grace W. K. Ho, Yim Wah Mak.

**Writing – review & editing:** Xuelin Zhang, Grace W. K. Ho, Yim Wah Mak.

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
