## [Decision Letter · Decision Letter 0]

12 Jun 2023

PONE-D-23-10500Effectiveness of a dyad-based acceptance and commitment therapy delivered via smartphone for chronic heart failure patients and their family caregivers: A study protocol for a randomized controlled trialPLOS ONE

Dear Dr. Mak,

Thank you for submitting your manuscript to PLOS ONE. After careful consideration, we feel that it has merit but does not fully meet PLOS ONE’s publication criteria as it currently stands. Therefore, we invite you to submit a revised version of the manuscript that addresses the points raised during the review process.

We look forward to receiving your revised manuscript.

Kind regards,

Cho Lee Wong, PhD

Academic Editor

PLOS ONE

Journal Requirements:

"NO authors have competing interests"

3. Please ensure that you include a title page within your main document. You should list all authors and all affiliations as per our author instructions and clearly indicate the corresponding author.

Additional Editor Comments:

The study addresses an important topic and the protocol is well-written. Please address reviewers' comments.

Reviewers' comments:

Reviewer's Responses to Questions

**Comments to the Author**

1. Does the manuscript provide a valid rationale for the proposed study, with clearly identified and justified research questions?

Reviewer #1: Yes

Reviewer #2: Yes

Reviewer #3: Yes

2. Is the protocol technically sound and planned in a manner that will lead to a meaningful outcome and allow testing the stated hypotheses?

Reviewer #1: Yes

Reviewer #2: Yes

Reviewer #3: Partly

3. Is the methodology feasible and described in sufficient detail to allow the work to be replicable?

Reviewer #1: Yes

Reviewer #2: Yes

Reviewer #3: No

4. Have the authors described where all data underlying the findings will be made available when the study is complete?

Reviewer #1: Yes

Reviewer #2: Yes

Reviewer #3: No

5. Is the manuscript presented in an intelligible fashion and written in standard English?

Reviewer #1: Yes

Reviewer #2: Yes

Reviewer #3: Yes

6. Review Comments to the Author

You may also provide optional suggestions and comments to authors that they might find helpful in planning their study.

Reviewer #1: Thanks for giving the opportunity to review the paper. The authors report a study protocol of ACT in chronic heart failure patients and their family caregivers. The topic is interesting and relevant to the journal, however, the manuscript has a number of issues that must be addressed before it can be considered further for publication. Specific comments are listed below.

Title:

1. Will the authors consider clarifying the primary outcome of the study in the title?

Introduction:

1. Lines 41-42. Could this sentence be more specific? What do patients anxious/depressive about? Any specific uncertainty experiences? Please also add citations for them.

2. It is suggested to add an optional definition for HRQOL.

3. Lines 51-52, how the psychological states are linked to HRQOL. Please specify their relationships.

4. As HRQOL is the primary outcome of the study, it could better to highlight the HRQOL of patients and caregivers and the impacts in an independent paragraph.

5. Please make sure that a thorough literature review of non-pharmacological interventions on dyads with chronic illness has been conducted with references citations. Please also be specific in summarising the limitations of current interventions on your target population. E.g. lines 66-68, why CBT and motivation enhancement are not sufficient to promote CHF self-management.

6. Line: it is unclear on ‘circumstances of context’

7. More than two studies have been conducted amongst patients with advanced cancer and their caregivers. Please make sure a thorough literature review was conducted again. Besides, why only review ACT studies on advanced cancer dyads? Any previous studies of ACT on other chronic illnesses that could be learned about?

8. It is suggested to highlight the necessity/significance to conduct an online-based ACT in your target population.

9. Line 83, ‘previous studies’ in which population?

Methods:

1. Lines 128-129, how 14 hours is identified? Any references?

2. Lines 147-148, ‘The authors report on a meta-analysis of ACT on psychological and quality of life outcomes in patients with advanced cancer.’ In this case, how could the authors ensure the study consistent with the planned frequency and duration of the intervention?

3. Line 186, please specify the ‘registered nurse’ from the author list, the principal investigator?

4. In the intervention section, the author said each session will be led by a registered nurse. In the intervention group, two registered nurse were described. Please be specific.

5. Lines 212-213, could authors specify the details of the revisions on Version 1 protocol?

6. Lines 243-244, as the second facilitator has a bachelor’s degree and only received two-day ACT training, which is different from the first facilitator who is a doctoral student with 28-day training, how could the author make sure this facilitator is able to independently deliver the interventions to participants?

7. Why patient’s HRQOL will be evaluated by two different scales?

8. Has the protocol been registered? If Yes, please provide the registered number.

Reviewer #2: The study is of interest and described in depth. Its potential usefulness is rather clear, and it will be have an impact on future researches in the field. Some comments follow.

1. Firstly, I suggest to improve the data collection plan. Up to what it my knowledge, cardiopulmonary excercise testing improves the prediction of risks. There are a series of works, see https://www.scopus.com/authid/detail.uri?authorId=6603721046, showing interesting results. The MECKI score is a valuable reference.

2. I appreciate the use of GEE modelling. However, I am wondering if you could better justify such a choice. There is a wide range of alternatives available in the literature, and e.g. random effects modelling is often preferred. Moreover, please specify how time-dependence is taken into account.

3. Missingness could strongly bias the results. The methods should be extended to consider missing not at random mechanism.

4. I am wondering why survival models are not considered or, even better, joint models with longitudinal and survival modelling. The latter represents the state of the art in modelling data having the same structure as yours.

5. A crucial assumption in defining the outcome is that the items have all the same probability to arise. This is quite a strong assumption to be carefully checked. Multinomial regressions could be considered instead. Moreover, please, be aware that even considering the score you propose, the outcome is on a limited range [0;100] and this data feature must be properly accounted for. Maybe, a beta regression or something similar should be considered.

Reviewer #3: This protocol paper presents the rationale and design of a dyadic, group-based acceptance and commitment therapy (ACT) intervention for patients with chronic heart failure and their family caregivers. Strengths of this protocol include the high level of detail and focus on an understudied population with great need for these types of interventions. The following issues warrant attention:

1. In the abstract, it is not clear that it is a dyadic, group-based intervention. One cannot tell that the authors are referring to groups of dyads receiving the intervention.

2. Lines 81-82: the telephone modality is said to limit the “validity” of prior research. It is possible that the phone-based modality made the intervention less efficacious, but the term “validity” is inappropriate here. Larger trials and meta-analyses are warranted before concluding that phone-based interventions are less efficacious. In some trials, phone-based interventions have reduced distress in medical populations.

3. More rationale for the group-based format is needed. There are pros and cons to this approach.

4. In the introduction, more conceptual links could be provided between the ACT intervention and outcomes of this trial. All key terms, such as psychological flexibility, should be defined in the introduction.

5. Lines 147-148: How can sessions be suspended during hospitalization and then resumed once the patient returns home? Does the entire group not meet during the patient’s hospitalization? Is there a maximum time period before the dyad is withdrawn?

6. Are dyads allowed to attend make-up sessions if they miss a group session?

7. The facilitators have limited experience with ACT. Also, are there different interventionists for the two study conditions?

8. Lines 179, 292-296: are the outcome assessments conducted in-person, via phone, online or via a combination of formats? There appears to be inconsistencies throughout the paper.

9. The power analysis needs to take into account the clustering of dyads within groups. Also, the statistical plan needs to account for the non-independent nature of the data. For example, mixed-effects modeling for dyadic data could be used for outcome measures completed by both patients and caregivers.

10. The entire manuscript needs to be edited for grammatical errors and awkward word usage. Also, the title for Figure 2 is missing words at the end.

7. PLOS authors have the option to publish the peer review history of their article (what does this mean?). If published, this will include your full peer review and any attached files.

Reviewer #1: No

Reviewer #2: No

Reviewer #3: No

---

## [Author Response · Author response to Decision Letter 0]

9 Aug 2023

Thank you for providing us with the opportunity to revise our paper. We have carefully edited the manuscript following the PLOS ONE author guidelines. We confirm that none of the authors have any competing interests. Additionally, we have added a title page containing the necessary information, including the title and all authors' details.

In response to the reviewers' feedback, we have thoroughly addressed their concerns and incorporated their comments and suggestions into the revised manuscript. Below is a summary of our response to the comments:

Comments Response (highlighted the revision made in the manuscript)

Reviewer #1: 

Thanks for giving the opportunity to review the paper. The authors report a study protocol of ACT in chronic heart failure patients and their family caregivers. The topic is interesting and relevant to the journal, however, the manuscript has a number of issues that must be addressed before it can be considered further for publication. Specific comments are listed below. 

Title:

1. Will the authors consider clarifying the primary outcome of the study in the title?

Response: 

Thank you for your comments, we included the primary outcomes of the study in the title, which is: 'Effectiveness of a videoconferencing group-based dyad acceptance and commitment therapy on the quality of life of chronic heart failure patients and their family caregivers: A study protocol for a randomized controlled trial’ in line 2-3 on page 1.

Introduction:

1. Lines 41-42. Could this sentence be more specific? What do patients anxious/depressive about? Any specific uncertainty experiences? Please also add citations for them.

Response: 

Thank you for your comments, we added more specific information related to individual experience contributing their psychological distress and problematic behaviors, which can be found in line 62-70 on page 4-5.

''However, it is important to recognize that engaging in CHF self-care activities, such as taking note of food items or seeking professional help, can have the effect of causing the individual to establish a psychological connection to the tangible and potentially distressing consequences associated with CHF, thereby evoking thoughts about the illness and eliciting reactions to its potential dangers. For instance, CHF patients may relate their interactions with healthcare providers to their "sick identity". This may lead them to consciously or unconsciously choose to avoid seeking professional help as a way to deny their CHF disease and cope with feelings of abnormality (21). These attempts to avoid or control their inner experiences can contribute to the development of psychological symptoms and lead to avoidance behaviors (22, 23, 24, 25, 26). Studies have shown high prevalence rates of anxiety (55.5%) and depression (ranging from 22% to 42%) among CHF patients (27, 28). Similarly, FCs also experience elevated levels of anxiety (50%) (29), depression (23-47%) (30), and a high level of perceived caregiving burden (31). Engaging in avoidance behaviors, such as socially isolating oneself (32, 33), being reluctant to seek support (34), being excessively vigilant in providing care to one’s patient (34), experiencing relationship issues (35, 36), eating salty food, being physically inactive, and not adhering to prescribed medicines (37, 38, 39), can be observed as consequences of the psychological distress experienced by both CHF patients and FCs.''

2. It is suggested to add an optional definition for HRQOL.

Response: 

Thank you for your comments, we added definition for quality of time, which can be found in line 50-51 on page 4.

 ''Quality of life (QOL) is a multidimensional construct that reflects an individual perception of their physical and mental health (7).''

3. Lines 51-52, how the psychological states are linked to HRQOL. Please specify their relationships.

Response: 

Thank you for your comments, we added the relationship of psychological states and QOL, which can be found in line 78-81 on page 5.

''These emotional and behavioral responses exhibited by each member within a dyad can have a reciprocal influence on the other (40, 41), resulting in a decrease in the quality of their relationship (42), poorer physical functioning (43), and reduced QOL (36, 44) for both patients and their FCs.''

4. As HRQOL is the primary outcome of the study, it could better to highlight the HRQOL of patients and caregivers and the impacts in an independent paragraph.

Response: 

Thank you for your comments, we highlighted the QOL and its impact on patient and caregiver outcomes, which can be found in line 50-58 on page 4.

 ''Quality of life (QOL) is a multidimensional construct that reflects an individual perception of their physical and mental health (7). Patients with CHF have generally lower QOL than healthy individuals and those with other chronic conditions (8, 9, 10). The reduced QOL in CHF patients increases their risk of rehospitalization and mortality (11). It is important to note that the FCs of CHF patients also experience impaired QOL to a degree, comparable to that of the patients themselves (12). A decline in the health status of FCs contributes to an increased risk of developing morbidities such as hypertension and cardiovascular disease (13), and may even lead to premature mortality (14, 15). Moreover, the diminished health status of FCs is associated with poorer health outcomes for CHF patients due to the compromised quality of the care provided by FCs (16, 17).''

5. Please make sure that a thorough literature review of non-pharmacological interventions on dyads with chronic illness has been conducted with references citations. Please also be specific in summarising the limitations of current interventions on your target population. E.g. lines 66-68, why CBT and motivation enhancement are not sufficient to promote CHF self-management.

Response: 

Thank you for your comments, we added more information on non-pharmacological interventions on dyads with chronic illness and the limitations of current interventions for CHF condition, which can be found in line 83-121 on page 5-7.

''Non-pharmacological interventions for CHF often include CHF education, self-care skill training, and psychosocial interventions; however, these interventions have shown limited success in patient outcomes (45, 46). Possible reasons for this lack of efficacy are the failure to recognize the efforts of family members and address the caregiving context (47). Similarly, the review identified limited evidence of the beneficial effects of interventions targeting CHF FCs (48, 49). Within the broader context of chronic illness, research has shown that interventions targeting both patients and FCs together, known as dyad interventions, are more effective in improving outcomes for both individuals compared to interventions focused solely on patients or FCs separately (50, 51, 52, 53). A recent meta-analysis of 13 randomized controlled trials (RCTs) published from 1992 to 2017 examined the effectiveness of dyad interventions focusing on chronic conditions such as cancer, chronic obstructive pulmonary disease, and rheumatoid arthritis (53). The findings of the meta-analysis demonstrated that dyad interventions had significant positive effects on both patient and family caregiver outcomes. Specifically, dyad interventions were found to be more effective in improving patient outcomes (Cohen’s d = 0.34), including physical symptoms, psychosocial functioning, and the patient-FC relationship, compared to interventions that targeted only the patients. Furthermore, family caregiver outcomes, such as anxiety, depression, and the patient-FC relationship, also showed significant improvements (d = 0.68) with dyad interventions. It should be noted that a considerable percentage of the included studies (39%) utilized a cognitive-behavioral therapy (CBT) approach, which was identified as one of the most effective in achieving positive outcomes for patients within the dyad interventions. 

Yet, despite the recent recognition of the importance of involving FCs in CHF interventions, the available evidence from a recent systematic review (54) indicates that only a limited number of experimental studies, specifically three out of 12 trials, focused on providing non-pharmacological support to optimize QOL and related outcomes for CHF patients and their FCs (55, 56, 57). These studies shed light on the challenges experienced by both patients and FCs in the context of CHF. However, because health outcomes vary across studies, the impact of these interventions on QOL and other psychological outcomes remains inconclusive (55, 56, 57). The main components of the interventions that were reviewed included CHF education and psychological support, with one study employing a CBT approach (56). The aim of implementing this intervention was to identify and correct CHF self-care related thoughts through cognitive efforts for both patients and their FCs. This individual study found positive effects on the mental dimension of QOL for both patients and caregivers, as well as on the level of depression among FCs post-intervention and at the nine-month follow-up (56). However, these effects did not reach the level of statistical significance compared to the usual care control (56). While CBT is a widely used psychological intervention for addressing psychological issues related to chronic physical conditions, it been criticized for its limited efficacy and low reproducibility (58). It is important to consider that attempting to modify difficult thoughts alone may have a limited effect on bringing about long-term beneficial changes in the psychological states of distressed individuals. This is because the efforts of individuals to alter their thoughts may not fully address the broader social and material influences that shape their experiences (59, 60, 61).''

6. Line: it is unclear on ‘circumstances of context’

Response:

Thank you for your comments. We have reorganized the paragraph, and the specific comments you mentioned are no longer applicable. 

7. More than two studies have been conducted amongst patients with advanced cancer and their caregivers. Please make sure a thorough literature review was conducted again. Besides, why only review ACT studies on advanced cancer dyads? Any previous studies of ACT on other chronic illnesses that could be learned about?

Response: 

Thank you for your comments, clarification regarding these issues was made in line 138 to 149 on page 8.

''However, it is worth noting that a significant percentage of existing studies on ACT primarily focus on patients or FCs attending ACT interventions independently. One systematic review of 24 studies published from 2014 to 2020 evaluated the use of ACT interventions for family caregivers providing care for a person with a long-term care condition (66). Only one of those studies—a pilot study—included adult patients accompanied by their corresponding FCs in the ACT intervention (67). One recent pilot study has also explored effects of dyad-based ACT interventions (68). These two pilot studies adopted ACT interventions for dyads with a patient with lung cancer (67) or gastrointestinal cancer (68), respectively. Both interventions were delivered via telephone in an individual/dyad format. Although no statistically significant differences were found in QOL, use of health services, and caregiver distress after the intervention when compared to those who received the usual care, high completion rates and levels of satisfaction among those participating in these studies provide early support for the use of a remote ACT intervention for patient-FC dyads.''

8. It is suggested to highlight the necessity/significance to conduct an online-based ACT in your target population.

Response: 

Thank you for your comments. We have addressed the necessity to use a videoconferencing group format in this study, which can be found in lines 150-163 on pages 8-9.

''In the context of patient-FC dyad interventions, the requirement for in-person participation can pose challenges to scalability and accessibility (54), particularly in the presence of transportation difficulties (69). Remote interventions, such as those delivered through telephone and video, have emerged as promising alternatives. A previous review on mental health care has shown that both telephone and video interventions are not inferior to in-person formats and can effectively improve psychological outcomes (70). However, videos can provide important visual information that telephone interventions cannot, allowing for the observation of non-verbal cues that may be relevant to psychological status and enhancing the therapeutic relationship between participants and interventionists (70). In addition, while the effects of ACT delivered in a group format have been found to be equivalent to individual-based interventions (71), the group format offers unique therapeutic benefits. These benefits include reducing feelings of isolation, facilitating shared learning from others' experiences, and providing opportunities for modeling new coping strategies and behaviors (72). Therefore, a videoconferencing group format was chosen as the mode of delivery for this intervention.''

9. Line 83, ‘previous studies’ in which population?

Response: 

Thank you for your comments. We have reorganized the paragraph, and the specific comments you mentioned are no longer applicable.

Methods:

1. Lines 128-129, how 14 hours is identified? Any references?

Response: 

Thank you for your comments. We have revised the sentence to ' who would be the person with the highest average number of contact hours among their family members ' in lines 200-201 on page 10. 

2. Lines 147-148, ‘The authors report on a meta-analysis of ACT on psychological and quality of life outcomes in patients with advanced cancer.’ In this case, how could the authors ensure the study consistent with the planned frequency and duration of the intervention?

Response: 

Thank you for your comments. We added information to support the rationale of the intervention frequency and duration in lines 297-306 on page 15.

''A four-session group-based ACT intervention has been deemed adequate for patients and their FCs based on previous evidence from systematic reviews (63, 104). For patients with chronic diseases, a systematic review found that ACT interventions consisting of less than five group sessions demonstrated significant medium-to-large improvements in disease management compared to the usual care (63). These improvements were observed in various health indicators, such as QOL, disease self-care behavior, and physical functioning, in patients with epilepsy, cancer, diabetes, and cardiac conditions (63). As for the FCs of patients with long-term illnesses, they showed significant improvement in various health outcomes after two to four group sessions of an ACT intervention when compared with the usual care, as indicated in a recent systematic review (66). These outcomes included: anxiety, depression, QOL, and family functioning (66).''

3. Line 186, please specify the ‘registered nurse’ from the author list, the principal investigator?

Response: 

Thank you for your comments. We added the initial of the registered nurses to line 337, 339, and 365 on page 16-17.

4. In the intervention section, the author said each session will be led by a registered nurse. In the intervention group, two registered nurse were described. Please be specific.

Response: 

Thank you for your comments. To avoid confusion, we deleted the statement regarding the specific number of registered nurses leading each session in the intervention section, and we added the relevant information in line 336 and 364 on page 16 and 17.

5. Lines 212-213, could authors specify the details of the revisions on Version 1 protocol?

Response: 

Thank you for your comments, we add the detailed information regarding the revision on version 1 protocol in line 293 to 296 on page 14.

 ''Several adaptations were made, such as emphasizing the interdependent nature of the dyadic relationship, adopting ACT experiential exercises to accommodate patient-FC dyads rather than individual patients or caregivers, and utilizing in-session dyadic techniques such as role-play and mutual elicitation of feedback. ''

6. Lines 243-244, as the second facilitator has a bachelor’s degree and only received two-day ACT training, which is different from the first facilitator who is a doctoral student with 28-day training, how could the author make sure this facilitator is able to independently deliver the interventions to participants?

Response: 

Thank you for your comments, we clarified this issue in line 335 to 337 and 339 to 342 on page 16.

 ''Due to the experiential learning and load of the online course material, the ACT-based intervention will be delivered to the patient-family caregiver dyads by two registered nurses working together. 

 …

The co-facilitator (CX) is a registered nurse at a local hospital who holds a bachelor's degree in nursing and has received two days of introductory-level training in ACT led by ACT experts in China. This training equipped her with the foundational knowledge and skills necessary to assist in delivering the online ACT intervention.''

7. Why patient’s HRQOL will be evaluated by two different scales?

Response: 

Thank you for your comments, we clarified this issue in line 386 to 394 on page 21.

 ''Generic measures, such as the EuroQol five-dimensional five-level (EQ-5D-5L) scale and the EuroQol visual analog scale (EQ-VAS) (98), will provide a broad evaluation of QOL across different populations and medical conditions. Disease-specific measures, such as the short form of the Kansas City Cardiomyopathy Questionnaire (KCCQ) (108), can be used to capture CHF-related issues that are particularly important to patients with CHF, and that are often sensitive to change (109). The evaluation of the patients' QOL will be conducted using different scales: the KCCQ (108), EQ-5D-5L, and EQ-VAS (98). These measures provide a comprehensive assessment of patient QOL from different perspectives (109). ''

8. Has the protocol been registered? If Yes, please provide the registered number.

Response: 

Thank you for your comments, we added the relevant information in line 549 to 550 on page 28.

 ''The study protocol was registered in ClinicalTrials.gov (Identifier: NCT04917159; Registered on 08-Jun-2021).''

Reviewer #2: 

The study is of interest and described in depth. Its potential usefulness is rather clear, and it will be have an impact on future researches in the field. Some comments follow.

Response: 

Thank you for your positive response.

1. Firstly, I suggest to improve the data collection plan. Up to what it my knowledge, cardiopulmonary excercise testing improves the prediction of risks. There are a series of works, see https://www.scopus.com/authid/detail.uri?authorId=6603721046, showing interesting results. The MECKI score is a valuable reference.

Response: 

Thank you for your comments. We revised it accordingly. Please see line 392-393 on page 21. 

 ''They will collect data through telephone interviews of patients with CHF and their FCs.''

2. I appreciate the use of GEE modelling. However, I am wondering if you could better justify such a choice. There is a wide range of alternatives available in the literature, and e.g. random effects modelling is often preferred. Moreover, please specify how time-dependence is taken into account.

Response:

Thank you for your comments, we changed our data analysis plan. Multilevel modeling will be used in this study. Please see line 519-545 on page 28-29. 

 ''Data will be analyzed following the intention-to-treat principle. To address missing data, multiple imputations will be employed with 50 imputed samples (125). For the outcomes variables reported by patients and FCs, multilevel modeling (MLM) is recommended to account for the nonindependence of data within the dyad and the presence of repeated measures (126). In cases where health outcomes apply exclusively to either the CHF patient or their FC, the analysis will employ individual MLM, which will involve interaction effects and the main effects of the study group and time. Assessment time points (baseline, immediately after the intervention, and three months post-intervention) are treated as categorical variables. Response variables that follow a normal distribution will be tested using the multilevel normal model with a linear link function. With response variables that display skewness in their distribution, the multilevel Poisson model with a log link function will be employed. To healthcare services utilization measures (e.g., the number of hospitalizations and emergency department visits), which are ordinal and involve count data, the multilevel Poisson model with a log link function will be applied. For outcomes that apply to both CHF patients and their FCs, MLM for dyadic data will be used (126). In the dyadic models, the fixed-effects parameters encompass the main effects of the study group, time, and role (patient vs. FC), along with their two- and three-way interaction effects. The significance of the group-by-time interaction effect will be used as evidence of the effects of the intervention. The three-way interaction among the study group, time, and role will be examined to assess the differential effects of the intervention on patients and FCs. The random-effects parameters account for separate residual variances for CHF patients and FCs, and the covariance between the residuals reflect the similarity in scores between the two members within the dyad at a specific time point while accounting for the fixed effects. Random intercepts for dyads will be incorporated into the variance of the model in the average outcome across patient-caregiver dyads. A significance level of p < 0.05 (two-tailed) will be utilized to determine statistical significance. A partial correlation coefficient (pr) will be computed as the effect size measure for each fixed effect (127). Among participants who complete the questionnaire, Cohen's d will be computed to assess the between-group effect for primary and secondary outcomes. In addition, subgroup analysis will be adopted for specific factors (e.g., intervention completion and non-completion).''

3. Missingness could strongly bias the results. The methods should be extended to consider missing not at random mechanism.

Response: 

Thank you for your comments, we revised it accordingly. Please see line 520 on page 28.

''To address missing data, multiple imputations will be employed with 50 imputed samples (125).''

4. I am wondering why survival models are not considered or, even better, joint models with longitudinal and survival modelling. The latter represents the state of the art in modelling data having the same structure as yours.

Response: 

Thank you for your comments. 

 ''In this study, the primary interest lies in evaluating the effects of ACT-based intervention on QOL and other multiple health outcomes for participants over the study period. While survival models and joint models with longitudinal and survival modelling can be valuable when investigating the effects of health determinants on trajectory of specific outcomes when dealing with longitudinal and time-to-event data (1,2), they are not the primary interest in this study. Given the specific study objectives and the nature of the dyad-focused intervention, it is more appropriate to use multilevel modeling (MLM). MLM can effectively account for the nonindependence of data within the dyad and handle repeated measures. This analytical approach allows for a comprehensive understanding of how the ACT-based intervention impacts various health outcomes for both patients and family caregivers. ''

References:

1. Fleming TR, Lin DY. Survival analysis in clinical trials: past developments and future directions. Biometrics. 2000;56(4):971-83.

2. Ibrahim JG, Chu H, Chen LM. Basic concepts and methods for joint models of longitudinal and survival data. J Clin Oncol. 2010;28(16):2796-801.

5. A crucial assumption in defining the outcome is that the items have all the same probability to arise. This is quite a strong assumption to be carefully checked. Multinomial regressions could be considered instead. Moreover, please, be aware that even considering the score you propose, the outcome is on a limited range [0;100] and this data feature must be properly accounted for. Maybe, a beta regression or something similar should be considered.

Response: 

Thank you for your comments. We revised accordingly. Please see line 526-532 on page 28. 

 ''Response variables that follow a normal distribution will be tested using the multilevel normal model with a linear link function. With response variables that display skewness in their distribution, the multilevel Poisson model with a log link function will be employed. To healthcare services utilization measures (e.g., the number of hospitalizations and emergency department visits), which are ordinal and involve count data, the multilevel Poisson model with a log link function will be applied. For outcomes that apply to both CHF patients and their FCs, MLM for dyadic data will be used (126).''

Reviewer #3: 

This protocol paper presents the rationale and design of a dyadic, group-based acceptance and commitment therapy (ACT) intervention for patients with chronic heart failure and their family caregivers. Strengths of this protocol include the high level of detail and focus on an understudied population with great need for these types of interventions. The following issues warrant attention:

Response: 

Thank you for your positive response. 

1. In the abstract, it is not clear that it is a dyadic, group-based intervention. One cannot tell that the authors are referring to groups of dyads receiving the intervention.

Response: 

Thank you for your comments. We revised the title accordingly, which can be found in line 2-3 on page 1. 

 ''Effectiveness of a videoconferencing group-based dyad acceptance and commitment therapy on the quality of life of chronic heart failure patients and their family caregivers: A study protocol for a randomized controlled trial.''

2. Lines 81-82: the telephone modality is said to limit the “validity” of prior research. It is possible that the phone-based modality made the intervention less efficacious, but the term “validity” is inappropriate here. Larger trials and meta-analyses are warranted before concluding that phone-based interventions are less efficacious. In some trials, phone-based interventions have reduced distress in medical populations.

Response:

 Thank you for your comments. We corrected these statements, which can be found in line 150-157 on page 8.

 ''Remote interventions, such as those delivered through telephone and video, have emerged as promising alternatives. A previous review on mental health care has shown that both telephone and video interventions are not inferior to in-person formats and can effectively improve psychological outcomes (70). However, videos can provide important visual information that telephone interventions cannot, allowing for the observation of non-verbal cues that may be relevant to psychological status and enhancing the therapeutic relationship between participants and interventionists (70).''

3. More rationale for the group-based format is needed. There are pros and cons to this approach.

Response: 

Thank you for your comments. We add the rational for group-based format in line 158-163 on page 8-9.

 ''In addition, while the effects of ACT delivered in a group format have been found to be equivalent to individual-based interventions (71), the group format offers unique therapeutic benefits. These benefits include reducing feelings of isolation, facilitating shared learning from others' experiences, and providing opportunities for modeling new coping strategies and behaviors (72). Therefore, a videoconferencing group format was chosen as the mode of delivery for this intervention.''

4. In the introduction, more conceptual links could be provided between the ACT intervention and outcomes of this trial. All key terms, such as psychological flexibility, should be defined in the introduction.

Response: 

Thank you for your comments. We revised it accordingly, which can be found in line 122 to 137 on page 7-8.

''Acceptance and Commitment Therapy (ACT), a transdiagnostic therapy grounded in relational frame theory and functional contextualism (62), appears to be well-suited for patients with CHF and their FCs in the context of home care. The aim of ACT is to improve daily functioning and QOL by cultivating psychological flexibility (63, 64). Psychological flexibility refers to the capacity to recognize situational demands at the present moment and to engage in value-driven actions despite external and internal barriers (62). ACT is about emphasizing the psychological, situational, and social contexts that modulate the behavioral influence of thoughts and emotions (62). Instead of attempting to alter cognitive content, the aim of ACT is to change the functions of problematic behavior to allow individuals to behave in line with their values (62). Through ACT, individuals are guided to accept and embrace their feelings, enabling them to detach themselves from the specific content of their thoughts by fostering mindful awareness of the thinking process, and encouraging them to align their actions with their personal values (62). A growing body of evidence suggests that ACT is effective in reducing anxiety and depressive symptoms and in improving the self-care and QOL of patients with cardiovascular diseases (CVD), including those with CHF (65). ACT has also been found to be beneficial for FCs who provide long-term care to patients, as it can help reduce their anxiety and depressive symptoms and improve their QOL (66).''

5. Lines 147-148: How can sessions be suspended during hospitalization and then resumed once the patient returns home? Does the entire group not meet during the patient’s hospitalization? Is there a maximum time period before the dyad is withdrawn?

6. Are dyads allowed to attend make-up sessions if they miss a group session?

Response: 

Thank you for your comments. We revised it accordingly, which can be found in line 269-275 on page 13-14.

''Patients and their FCs will be required to participate in the group sessions on a dyad basis. During the intervention period, if either member of the dyad has been hospitalized, both will be asked to suspend their participation in the program. Hospitalization typically requires individuals to prioritize their acute medical needs. After being discharged from the hospital and returning home, the patient-caregiver dyads will be asked to resume their participation within one week after discharge. If patient-FC dyads miss a session, a make-up session will be provided via videoconferencing within three days before the start of the next scheduled session. ''

7. The facilitators have limited experience with ACT. Also, are there different interventionists for the two study conditions?

Response: 

Thank you for your comments. We updated the facilitator training experience in line 335- 350 on page 16-17. In addition, because this is an assessor-blinded study, ensuring the blinding of the outcome assessor is important. The interventions will be implemented during the same time slot for both study groups. Thus, different interventionists will be assigned for each study condition. All interventionists have a minimum of three-year working experience on cardiac care. But two nurses will be assigned in the ACT-based intervention group due to the experiential learning and load of online course material, while one nurse will be designated to administer the intervention for the control group. The clarification can be found in line 335 to 336 and line 364 to 365 on page 16-17.

 ''Due to the experiential learning and load of the online course material, the ACT-based intervention will be delivered to the patient-family caregiver dyads by two registered nurses together. The primary facilitator (XZ) is also a doctoral student in nursing who has completed a total of 20 days of ACT workshops and a guided online 21-day ACT action camp led by ACT experts worldwide and in China. The co-facilitator (CX) is a registered nurse at the local hospital who holds a bachelor's degree in nursing, she has received two days of introductory-level training in ACT led by ACT experts in China. This training equipped her with the foundational knowledge and skills necessary to assist in delivering the online ACT intervention. 

Both of them have a minimum of three years of experience working with cardiac inpatients and have prior experience in co-facilitating group ACT-based interventions for patient-caregiver dyads during the pilot studies, where they conducted approximately 12 group sessions (a total of 24 hours) together. All the sessions in these pilot studies were video recorded for the purpose of quality assurance and fidelity checking. These recordings were then reviewed and discussed with an experienced ACT researcher (YWM). To further enhance their expertise in delivering the ACT-based intervention for the current study, the facilitators underwent role-play practices by using the finalized version of the intervention protocol for this study before the commencement of this study under supervision. 

…

The sessions for the CHF education control will be delivered by one registered nurse (YJ) who holds a master’s degree in nursing and has at least three years of experience working with cardiac inpatients.''

8. Lines 179, 292-296: are the outcome assessments conducted in-person, via phone, online or via a combination of formats? There appears to be inconsistencies throughout the paper.

Response: 

Thank you for your comments, we corrected these statements please see line 380 -382 on page 21.

''They will collect data through telephone interviews of patients with CHF and their FCs.''

9. The power analysis needs to take into account the clustering of dyads within groups. 

Response: 

Thank you for your comments. Conducting power calculations for the analyses of QOL (primary outcomes) on the dyad level for CHF patients is difficult due to the lack of detailed information on parameters from previous similar studies in the field of psychological intervention for dyads of patients and family caregivers, as well as the ACT intervention for dyads of patient and family caregiver. The clarification was provided in the line 219-222 on page 11.

''Power calculations necessitate specific information on health outcome indicators from previous studies (81). However, due to the unavailability of detailed dyadic information specifically for dyadic analysis on QOL and other health outcomes for CHF patients and their FCs, the prior power analysis conducted using G*Power in this study relied on individual-level data.

Also, the statistical plan needs to account for the non-independent nature of the data. For example, mixed-effects modeling for dyadic data could be used for outcome measures completed by both patients and caregivers.

Response: 

Thank you for your comments. The statistic plan was revised according, which can be found in lines 519-545 on page 28. 

''Data will be analyzed following the intention-to-treat principle. To address missing data, multiple imputations will be employed with 50 imputed samples (125). For the outcomes variables reported by patients and FCs, multilevel modeling (MLM) is recommended to account for the nonindependence of data within the dyad and the presence of repeated measures (126). In cases where health outcomes apply exclusively to either the CHF patient or their FC, the analysis will employ individual MLM, which will involve interaction effects and the main effects of the study group and time. Assessment time points (baseline, immediately after the intervention, and three months post-intervention) are treated as categorical variables. Response variables that follow a normal distribution will be tested using the multilevel normal model with a linear link function. With response variables that display skewness in their distribution, the multilevel Poisson model with a log link function will be employed. To healthcare services utilization measures (e.g., the number of hospitalizations and emergency department visits), which are ordinal and involve count data, the multilevel Poisson model with a log link function will be applied. For outcomes that apply to both CHF patients and their FCs, MLM for dyadic data will be used (126). In the dyadic models, the fixed-effects parameters encompass the main effects of the study group, time, and role (patient vs. FC), along with their two- and three-way interaction effects. The significance of the group-by-time interaction effect will be used as evidence of the effects of the intervention. The three-way interaction among the study group, time, and role will be examined to assess the differential effects of the intervention on patients and FCs. The random-effects parameters account for separate residual variances for CHF patients and FCs, and the covariance between the residuals reflect the similarity in scores between the two members within the dyad at a specific time point while accounting for the fixed effects. Random intercepts for dyads will be incorporated into the variance of the model in the average outcome across patient-caregiver dyads. A significance level of p < 0.05 (two-tailed) will be utilized to determine statistical significance. A partial correlation coefficient (pr) will be computed as the effect size measure for each fixed effect (127). Among participants who complete the questionnaire, Cohen's d will be computed to assess the between-group effect for primary and secondary outcomes. In addition, subgroup analysis will be adopted for specific factors (e.g., intervention completion and non-completion).''

10. The entire manuscript needs to be edited for grammatical errors and awkward word usage. Also, the title for Figure 2 is missing words at the end.

Response: 

Thank you for your comments, we have carefully reviewed the entire manuscript and had it professionally edited by an English editor. 

We believe that the revised manuscript now addresses all of the concerns raised by the reviewers and has been significantly improved as a result. 

Thank you again for your time and consideration.

---

## [Decision Letter · Decision Letter 1]

21 Nov 2023

PONE-D-23-10500R1Effectiveness of a videoconferencing group-based dyad acceptance and commitment therapy on the quality of life of chronic heart failure patients and their family caregivers: A study protocol for a randomized controlled trialPLOS ONE

Dear Dr. Mak,

Thank you for submitting your manuscript to PLOS ONE. After careful consideration, we feel that it has merit but does not fully meet PLOS ONE’s publication criteria as it currently stands. Therefore, we invite you to submit a revised version of the manuscript that addresses the points raised during the review process.

We look forward to receiving your revised manuscript.

Kind regards,

Cho Lee Wong, PhD

Academic Editor

PLOS ONE

Journal Requirements:

Reviewers' comments:

Reviewer's Responses to Questions

**Comments to the Author**

1. Does the manuscript provide a valid rationale for the proposed study, with clearly identified and justified research questions?

Reviewer #1: Yes

2. Is the protocol technically sound and planned in a manner that will lead to a meaningful outcome and allow testing the stated hypotheses?

Reviewer #1: Yes

3. Is the methodology feasible and described in sufficient detail to allow the work to be replicable?

Reviewer #1: Yes

4. Have the authors described where all data underlying the findings will be made available when the study is complete?

Reviewer #1: Yes

5. Is the manuscript presented in an intelligible fashion and written in standard English?

Reviewer #1: Yes

6. Review Comments to the Author

You may also provide optional suggestions and comments to authors that they might find helpful in planning their study.

Reviewer #1: The paper has been improved; some minor comments are as below:

1. The authors stated, ‘These emotional and behavioral responses exhibited by each member within a dyad can have a reciprocal influence on the other (40, 41), resulting in a decrease in the quality of their relationship (42), poorer physical functioning (43), and reduced QOL (36, 44) for both patients and their FCs.’ Please be more specific on how emotional and behavioral responses of each member influence the other.

2. Please consider shorten the length of the introduction, as this part is a bit long.

3. Language editing is needed before resubmission.

7. PLOS authors have the option to publish the peer review history of their article (what does this mean?). If published, this will include your full peer review and any attached files.

Reviewer #1: No

---

## [Author Response · Author response to Decision Letter 1]

1 Dec 2023

Comments Response (highlighted the revision made in the manuscript)

Reviewer #1:

The paper has been improved; some minor comments are as below:

1. The authors stated, ‘These emotional and behavioral responses exhibited by each member within a dyad can have a reciprocal influence on the other (40, 41), resulting in a decrease in the quality of their relationship (42), poorer physical functioning (43), and reduced QOL (36, 44) for both patients and their FCs.’ Please be more specific on how emotional and behavioral responses of each member influence the other.

Response: Thank you for your comments. We have added the statement regarding the emotional and behavioral responses within the dyad as below.

However, engaging in CHF self-care activities, such as taking note of food items or seeking professional help, can have the effect of causing the individual to establish a psychological connection to the tangible and potentially distressing consequences associated with CHF, thereby evoking thoughts about the illness and eliciting reactions to its potential dangers. For instance, CHF patients may relate their interactions with healthcare providers to their "sick identity". This may lead them to choose to avoid seeking professional help as a way to deny their CHF disease and cope with feelings of abnormality (21). FCs often dedicate to providing competent and timely care (22). Many FCs respond with unconscious excessive vigilance in ensuring the well-being and comfort of their patients, driven by their fear of potential future losses (23, 24). This heightened vigilance may persist even when the patient’s health is relatively stable and care demands are low (25, 26).

2. Please consider shorten the length of the introduction, as this part is a bit long. 

Response: Thank you for your suggestion. We have shortened the introduction to 129 lines.

3. Language editing is needed before resubmission. 

Response: Thank you for your suggestion. The paper has been edited by a professional editor.

---

## [Decision Letter · Decision Letter 2]

22 Jan 2024

Effectiveness of a videoconferencing group-based dyad acceptance and commitment therapy on the quality of life of chronic heart failure patients and their family caregivers: A study protocol for a randomized controlled trial

PONE-D-23-10500R2

Dear Dr. Mak,

We’re pleased to inform you that your manuscript has been judged scientifically suitable for publication and will be formally accepted for publication once it meets all outstanding technical requirements.

Kind regards,

Cho Lee Wong, PhD

Academic Editor

PLOS ONE

Additional Editor Comments (optional):

Reviewers' comments:

Reviewer's Responses to Questions

**Comments to the Author**

1. Does the manuscript provide a valid rationale for the proposed study, with clearly identified and justified research questions?

Reviewer #1: Yes

2. Is the protocol technically sound and planned in a manner that will lead to a meaningful outcome and allow testing the stated hypotheses?

Reviewer #1: Yes

3. Is the methodology feasible and described in sufficient detail to allow the work to be replicable?

Reviewer #1: Yes

4. Have the authors described where all data underlying the findings will be made available when the study is complete?

Reviewer #1: Yes

5. Is the manuscript presented in an intelligible fashion and written in standard English?

Reviewer #1: Yes

6. Review Comments to the Author

You may also provide optional suggestions and comments to authors that they might find helpful in planning their study.

Reviewer #1: Thanks for the authors' effort on the paper. All comments have been addressed. I have no more comments on this paper.

7. PLOS authors have the option to publish the peer review history of their article (what does this mean?). If published, this will include your full peer review and any attached files.

Reviewer #1: No

---

## [Editor Report · Acceptance letter]

27 Mar 2024

PONE-D-23-10500R2 

PLOS ONE

Dear Dr. Mak, 

I'm pleased to inform you that your manuscript has been deemed suitable for publication in PLOS ONE. Congratulations! Your manuscript is now being handed over to our production team.

Kind regards, 

on behalf of

Dr. Cho Lee Wong 

Academic Editor

PLOS ONE